# Derhamnosylmaysin Inhibits Adipogenesis via Inhibiting Expression of PPARγ and C/EBPα in 3T3-L1 Cells

**DOI:** 10.3390/molecules27134232

**Published:** 2022-06-30

**Authors:** Hang-Hee Cho, Sun-Hee Jang, Chungkil Won, Chung-Hui Kim, Hong-Duck Kim, Tae Hoon Kim, Jae-Hyeon Cho

**Affiliations:** 1Institute of Animal Medicine, College of Veterinary Medicine, Gyeongsang National University, Jinju 52828, Korea; greatcho2000@nate.com (H.-H.C.); sunhee5321@naver.com (S.-H.J.); wonck@gnu.ac.kr (C.W.); kimchi3237@gmail.com (C.-H.K.); 2Department of Public Health, Division of Environmental Health Science, New York Medical College, Valhalla, NY 10595, USA; hongduck_kim@nymc.edu; 3Department of Food Science and Biotechnology, Daegu University, Gyungsan 38453, Korea

**Keywords:** adipogenesis, Akt, derhamnosylmaysin, lipogenesis, 3T3-L1 cells

## Abstract

We investigated the effects of derhamnosylmaysin (DM) on adipogenesis and lipid accumulation in 3T3-L1 adipocytes. Our data showed that DM inhibited lipid accumulation and adipocyte differentiation in 3T3-L1 cells. Treatment of 3T3-L1 adipocytes with DM decreased the expression of major transcription factors, such as sterol regulatory element-binding protein-1c (SREBP-1c), the CCAAT-enhancer-binding protein (CEBP) family, and peroxisome proliferator-activated receptor gamma (PPARγ), in the regulation of adipocyte differentiation. Moreover, the expression of their downstream target genes related to adipogenesis and lipogenesis, including adipocyte fatty acid-binding protein (aP2), lipoprotein lipase (LPL), stearyl-CoA-desaturase-1 (SCD-1), acetyl-CoA carboxylase (ACC), and fatty acid synthase (FAS), was also decreased by treatment with DM during adipogenesis. Additionally, DM attenuated insulin-stimulated phosphorylation of Akt. These results first demonstrated that DM inhibited adipogenesis and lipogenesis through downregulation of the key adipogenic transcription factors SREBP-1c, the CEBP family, and PPARγ and inactivation of the major adipogenesis signaling factor Akt, which is intermediated in insulin. These studies demonstrated that DM is a new bioactive compound for antiadipogenic reagents for controlling overweight and obesity.

## 1. Introduction

Obesity is a medical condition defined as a disproportionate amount of adipose tissue storage in the human body, which is caused by an imbalance between energy intake and expenditure [1]. Abnormal or excessive accumulation of body fat extends to serious consequences for health, which may cause metabolic malfunction and interventional effects on different organ health. Excessive fat deposition results in obesity, which occurs through enhanced adipogenesis in adipose tissue. Subsequently, the accumulation of excess lipids in adipocytes is highly associated with pathological disorders such as dyslipidemia, diabetes, blood pressure disorders, and obesity-associated metabolic disease [2].

During adipogenesis, fibroblast-like preadipocytes acquire the morphological and biochemical characteristics of adult adipocytes, and blocking the differentiation of preadipocytes into mature adipocytes can decrease total fat mass and adipose tissue. Thus, isolation of active anti-adipogenic compounds for controlling adipocyte differentiation is a novel therapeutic approach for metabolic diseases involving clustering of excess abdominal fat, high blood pressure, and metabolic disease [3].

Polyphenols are the main class of biologically active compounds naturally produced in many plants as secondary metabolites. They are a significant part of the human diet, and evidence suggests that the consumption of dietary phenolic compounds is closely associated with their ability to modulate healthy metabolic status. In a variety of studies, the beneficial effect of polyphenols on lipid metabolism was demonstrated to prevent metabolic syndrome by decreasing body fat, body weight, and blood glucose and by improving lipid metabolism [4,5,6]. There are strong indications that polyphenols may be attractive candidates for the prevention of several metabolic disorders through the modulation of pathways of glucose and lipid metabolism.

Derhamnosylmaysin (DM) is a single compound extracted from centipede grass extracts, which contain maysin derivatives such as luteoin-6-C-boivinopyranose, luteolin, isoorientin, and rhamnosylisoorientin [7,8]. DM containing its maysin derivatives showed high levels of free radical-scavenging activity in vitro biochemical assays using 2,2,1-diphenyl-1-picrylhydrazyl (DPPH)-radical scavenging activity [8,9]. In particular, DM is a potential pancreatic lipase inhibitor that represents a class of polyphenolic and flavonoid compounds that have significant antioxidant activities either in vitro or in vivo [7,10].

The 3T3-L1 fibroblasts are able to differentiate into fat-laden adipocytes in a span of approximately one week upon induction using fetal bovine serum (FBS), dexamethasone (DEX), 3-isobutyl-1-methylxanthine (IBMX), and insulin [11,12]. A mixture (DMI) of insulin, IBMX, and DEX was used to chemically induce the differentiation of 3T3-L1 cells into adipocytes. In particular, DEX and IBMX have been identified as direct inducers of genes responsible for the expression of CCAAT/enhancer-binding protein delta (C/EBPδ) and CCAAT/enhancer-binding protein beta (C/EBPβ), respectively [13,14]. Insulin is known to stimulate adipocyte differentiation and increase the lipid deposition of differentiated 3T3-L1 adipocytes. Insulin also activates the phosphoinositide-3-kinase (PI3K)/Akt signaling pathways to activate adipogenic effects, and insulin-PI3K-Akt signaling is a positive regulator of adipogenesis and promotes the differentiation of preadipocytes by increasing the expression of the C/EBP family and peroxisome proliferator-activated receptor-gamma (PPARγ) [3,15,16]. Then, the differentiation of preadipocytes to adipocytes involves a comprehensive network including transcription factors responsible for the expression of key proteins that induce mature adipocyte formation. The process of adipogenesis also involves changes in cell morphology, induction of insulin sensitivity, and changes in the secretory capacity of cells [17].

During the early stages of differentiation, there is high expression of C/EBPβ and C/EBPδ in response to hormonal induction. Several studies have indicated that CCAAT/enhancer-binding proteins (C/EBPs) regulate the induction of preadipocyte differentiation and the modulation of gene expression in fully differentiated adipocytes [13,14]. Transcription factors, PPARγ and CCAAT/enhancer-binding protein alpha (C/EBPα) play early catalytic roles in the adipogenic differentiation pathway, diminish during the late stages of differentiation, which was demonstrated in previous reports [18,19]. The coordination of PPARγ with C/EBP transcription factors maintains adipocyte gene expression and forms positive feedback to mutually regulate the target genes involved in adipogenesis. In addition, sterol regulatory-element binding proteins (SREBPs) are a family of transcription factors that regulate lipid and fatty acid synthesis and energy storage and act as intercellular signaling nodes of convergence/divergence [20,21].

The C/EBP family, PPARγ, and SREBP-1c are the main regulators of adipogenesis and have been shown to exhibit extensive overlap in their transcriptional targets. These transcription factors are critical for coordinated regulation depending on the stage of adipogenesis. These transcription factors regulate normal adipocyte differentiation and targeting molecules such as adipocyte fatty acid-binding protein (aP2), lipoprotein lipase (LPL), stearyl-CoA-desaturase-1 (SCD-1), acetyl-CoA carboxylase (ACC), and fatty acid synthase (FAS) [14,22].

In this study, we first studied the anti-adipogenic effect of DM in 3T3-L1 cells. The effects of DM on adipocyte differentiation in 3T3-L1 preadipocytes were investigated by measuring triglyceride (TG) accumulation as well as the expression levels of adipocyte marker genes and their target genes. Moreover, to understand the specific mechanisms of these effects, we examined whether Akt activation is critical for the anti-obesity function of DM.

## 2. Results

### 2.1. Effects of DM on Lipid Accumulation

The antiadipogenic potential of DM was first examined to assess lipid accumulation in 3T3-L1 adipocytes. The chemical structure of DM is shown in Figure 1A. The 3T3-L1 preadipocytes were continuously incubated in DMI medium (insulin, IBMX, and dexamethasone) with different concentrations (0, 1.1, and 2.2  μM) of DM. Cultured 3T3-L1 cells differentiated into adipocytes and accumulated lipid droplets in the cytoplasm. After 8 days, the differentiation level of 3T3-L1 preadipocytes into lipid-laden adipocytes was determined with Oil Red O staining for triglyceride accumulation as an indicator of the degree of adipogenesis and lipid droplet biogenesis. According to the results of Oil Red O staining, treatment with DM (1.1 and 2.2  μM) markedly reduced the amount of lipid droplet formation compared to that of differentiated adipocytes (Figure 1B). Quantitative analysis confirmed that treatment of 3T3-L1 cells with DM significantly decreased the intracellular TG content in 3T3-L1 adipocytes on day 4 or 8 of differentiation (Figure 1C). Notably, treatment with DM dose- and time-dependently suppressed the accumulation of intracellular triglycerides in 3T3-L1 cells.

### 2.2. Effects of DM on Cell Viability

To study whether 1.1 or 2.2 μM DM shows cytotoxic effects on cell viability in 3T3-L1 preadipocytes, an MTT assay was conducted. As shown in Figure 1D, at a concentration of 2.2 μM, DM had no significant inhibitory effects on cell viability and did not cause cytotoxicity in 3T3-L1 cells after 4 or 8 days of incubation (Figure 1D).

### 2.3. Effects of DM on the Expression of Adipogenic-Transcription Factors during Adipocyte Differentiation

To assess the mechanism of reduction in intracellular triglyceride content during adipocyte differentiation, we investigated the expression levels of C/EBPβ, PPARγ, and C/EBPα, major transcription factors for adipocyte differentiation. It is well documented that during the induction of 3T3-L1 adipocyte differentiation, PPARγ and C/EBPs are activated by an MDI mixture [13]. To investigate the inhibitory effects of DM, we purified total RNA from differentiated 3T3-L1 cells on day 4 and performed RT-PCR. Indeed, DM treatment induced a decrease in the expression of C/EBPβ mRNA. It was also found that mRNA expression of transcriptional factors both PPARγ and C/EBPα inhibited the adipogenesis-related cellular events. Taken together, regulation skewed to adipogenic factors, the mRNA levels of C/EBPβ, C/EBPα, and PPARγ were reduced by treating differentiated 3T3-L1 with DM (Figure 2A), suggesting that DM exerted a strong inhibitory effect on the differentiation of 3T3-L1 cells via insulin treatment. Next, we investigated the protein expression levels of the adipogenic regulator (i.e., C/EBPs and PPARγ) during adipocyte differentiation on lipid accumulation in 3T3-L1 adipocytes. During the course of adipocyte maturation, DMI treatment significantly increased the protein expression of C/EBPβ, PPARγ, and C/EBPα, key transcription factors for adipocyte differentiation in 3T3-L1 cells, while DM markedly reduced the protein expression of C/EBPβ and C/EBPα compared to DMI-treated controls (Figure 2B,C). In addition, consistent with the decrease in TG accumulation and C/EBPβ expression, DM significantly decreased PPARγ and SREBP-1c expression in a dose-dependent manner compared to that in differentiated adipocytes.

### 2.4. Effects of DM on the Expression of Adipocyte-Specific Genes in 3T3-L1 Adipocytes

To examine whether the reduced expression of the C/EBP family, PPARγ, and SREBP-1c further regulated the expression levels of their target genes, such as aP2, FAS, and LPL, we assessed their expression during 3T3-L1 differentiation. Although treatment with DMI increased the expression levels of FAS, aP2, and LPL, DM treatment significantly decreased the expression of adipogenesis-specific genes involved in lipid metabolism, FAS, aP2, and LPL compared to those in DMI-treated differentiated adipocytes (Figure 3A). The effects of DM on the expression levels of the lipogenic genes including ACC and SCD-1 were measured by Western blotting. Treatment with DM significantly down-regulated the expression of the lipogenic genes, ACC and SCD-1, compared to the fully differentiated 3T3-L1 adipocytes (Figure 3A). Together, these results indicated that DM reduced the expression of PPARγ, SREBP-1c, and C/EBPs and their target genes involved in adipogenesis and lipogenesis.

### 2.5. Effects of DM on Akt Phosphorylation in 3T3-L1 Adipocyte Differentiation

Akt is known to play a major role in glucose regulation and lipid metabolism in insulin signaling. Treatment with a mixture (DMI) of insulin, dexamethasone, and 3-isobutyl-1-methylxanthine induces the adipogenic differentiation process in 3T3-L1 cells. Insulin treatment of differentiated 3T3-L1 cells results in a significant increase in Akt activation associated with Akt phosphorylation. To determine the effect of Akt phosphorylation, we investigated whether DM downregulated the Akt phosphorylation level in differentiated 3T3-L1 cells. As shown in Figure 3C,D, the phosphorylation level of Akt was increased following the DMI-induced differentiation of 3T3-L1 cells, while DM treatment inhibited DMI-induced Akt phosphorylation (Ser473) in 3T3-L1 adipocytes. These data indicated that DM significantly decreased the phosphorylation of Akt, suggesting that DM inhibited the Akt signaling pathway involved in the major adipogenesis cascade in differentiating 3T3-L1 adipocytes.

We employed the chemical inhibitor of the Akt, LY294002 to confirm whether the inhibitory effect of DM in adipogenesis is dependent on Akt signaling. Differentiated 3T3-L1 cells induced by MDI media showed a significant increase in lipid accumulation, compared with non-differentiated 3T3-L1 cells (Figure 3E). However, treatment with 10 μM LY294002 significantly decreased the contents of TG in the differentiated 3T3-L1 cells. Moreover, we found that the inhibitory effects of DM on lipid droplet formation were further increased in the combination groups (LY29400 and DM) compared with DM alone (Figure 3E). These results indicated that the role of Akt signaling is significantly associated with the functional modulation of the DM, which reduces adipogenesis in 3T3-L1 cells.

## 3. Discussion

Obesity-related conditions are a public health concern due to their impact on metabolic disorders and mental disorders covering a wide age range depending on the specific type of disease tropism. Underlying obesity-related metabolic pathogenesis includes adipogenesis and differentiation, and adipocytes play an important role in the regulation of energy storage, fatty acid metabolism, and energy expenditure. Adipose tissue also has an important function in energy balance by regulating lipid metabolism and glucose homeostasis [23]. However, excessive fat deposition in adipose tissue is a fast-growing risk for numerous chronic diseases, such as heart disease, diabetes, and hypertension. Overweight-related obesity is no longer regarded as only a cosmetic problem, in that it is the result of the abnormal fat accumulation associated with an increased risk for the development of numerous adverse health conditions [24,25].

There is no doubt that preventing obesity plays an essential role in having good health. In general, obesity is related to the extent of preadipocyte differentiation into mature adipocytes, excessive intracellular lipid accumulation, and lipogenesis. Numerous polyphenolic compounds are potential agents for treating obesity, and DM also exhibits strong antioxidant activities as a flavonoid compound [7,9]. Of note, it was suggested adipocyte expansion was associated with adipogenic cellular events, which are a critical step for the onset of the metabolic alteration in end phenotypic outcomes such as obesity [3,26], following reflection of advanced studies by considering caveats of connectivity between adipogenesis and metabolic signaling.

In the present study, in the efforts of new exploration aimed at the relationship be-tween structural and activity in natural metabolites, we investigated the inhibitory effect of DM on adipocyte differentiation and lipid accumulation in 3T3-L1 preadipocytes. We characterized cellular and molecular signaling pathways focused on DM-driven anti-obesity utilizing mouse fibroblast 3T3-L1 preadipocytes. As a phenotypic indicator, the intracellular lipid accumulation was examined with Oil Red O staining for lipid droplets as an indicator of the degree of adipogenesis following exposure to inducers on the adipogenicity in 3T3-L1 cells. Microscopic images of 3T3-L1 cells with Oil Red O staining revealed that the accumulation of lipid droplets increased with adipogenic differentiation medium (DMI) treatment. However, our results showed that treatment with DM resulted in a decrease in the lipid droplet spot, which may reflect a lipid content decrease in Oil Red O staining, indicating that DM inhibited the differentiation of 3T3-L1 preadipocytes by decreasing lipid accumulation in the intracellular region.

To further study the inhibitory effects of DM on adipogenic differentiation, the mRNA levels of adipogenic biochemical markers such as C/EBPβ, C/EBPα, and PPARγ during the differentiation of 3T3-L1 cells into adipocytes were measured. When 3T3-L1 preadipocytes were differentiated with MDI treatment, the mRNA expression levels of adipogenesis transcription factors (C/EBPβ, C/EBPα, and PPARγ) increased compared to those of the untreated DMI control. However, DM treatment led to a significant attenuation by increasing the dose of DM at the mRNA levels of the C/EBPβ, SREBP-1, and PPARγ genes. These results indicated that the expression levels of C/EBPβ, PPARγ, and SREBP-1 mRNA were markedly inhibited by DM at the transcriptional level, suggesting that DM affected major regulators associated with signaling for adipocyte differentiation in 3T3-L1 cells.

To understand the mechanism underlying the anti-adipogenic effect of DM, we investigated the protein expression levels of major adipogenic genes related to lipid metabolism. The protein expression levels of the C/EBPβ, SREBP-1, C/EBPα, and PPARγ genes were significantly reduced with DM treatment compared with the control, which showed that the decrease was dose dependent. Together, DM inhibited the mRNA and protein expression of adipogenic transcription factors during the differentiation of 3T3-L1 preadipocytes into mature adipocytes. Adipogenesis involves the interaction and regulation between transcription factors, such as C/EBPs and PPARγ, which are key regulators of adipose cell or tissue development [13,16]. In particular, the expression of PPARγ and C/EBPα is sustained in differentiated adipocytes to promote lipid accumulation [27], which results in an adipocyte phenotype with significantly increased triglyceride levels. Moreover, the expression of SREBP-1c increased in adipocytes, which resulted in an increase in lipogenic gene expression broadly involved in coordinating lipid metabolism [3,21]. SREBP-1c also regulates the expression of genes related to cholesterol metabolism and fatty acid desaturation [21]. The results presented in this study showed that DM treatment strongly reduced the expression levels of the C/EBP family, SREBP-1c, and PPARγ compared with those in differentiated control cells. Consequently, these data demonstrated that DM inhibited adipogenesis via suppression of adipogenic transcription factors and lipid accumulation gene expression in adipocytes.

Consistent with the suppression of adipogenic major factors, the expression of lipogenesis-related genes such as FAS, SCD-1, and ACC was also inhibited by DM treatment. In addition, DM treatment resulted in decreased expression of fatty acid transport-related genes, LPL, and aP2 in 3T3-L1 adipocytes compared to vehicle-control cells. C/EBPα induces adipogenesis through PPARγ, which accelerates the activation of adipocyte-specific genes related to lipid metabolism and activates the expression of lipid-metabolizing enzymes, such as aP2, LPL, and FAS [14]. The expression of SREBP-1c is induced by insulin stimulation in adipocytes, and SREBP-1c increases the expression of lipogenic genes involved in lipogeneses, such as ACC, FAS, and SCD-1 [28]. In this study, the expression of the C/EBP family, SREBP-1c, and PPARγ target genes aP2, LPL, FAS, and SCD-1 was decreased by DM treatment in 3T3-L1 adipocytes. These results indicated that DM downregulated the primary regulator of preadipocyte differentiation into mature adipocytes and resulted in the suppression of their downstream genes that promote preadipocyte differentiation into mature adipocytes. Therefore, our data demonstrated that DM significantly suppressed adipocyte differentiation and lipid accumulation in 3T3-L1 adipocytes through the inhibition of adipogenesis-related genes and the inactivation of lipogenesis-related genes in mature adipocytes.

Moreover, we investigated whether Akt was involved in DM-inhibited 3T3-L1 adipocyte differentiation. Akt induces numerous intracellular signaling pathways that play a central role in various cellular processes, such as cell proliferation and differentiation. Akt affects lipid metabolism in adipogenesis through the insulin pathway, and activation of the Akt pathway in 3T3-L1 preadipocytes contributes to adipocyte differentiation [29,30]. Moreover, lack of or inhibition of Akt activation leads to impairment of adipogenesis and lipogenesis in adipose tissue, whereas Akt overexpression in Akt-deficient cells enhances the induction of adipose cell lipogenesis [31]. It was also well reported that the Akt/insulin signaling pathway enhances 3T3-L1 preadipocyte differentiation by regulating PPARγ and C/EBPα [29,32]. The expression of Akt induces an important association between the PI3-kinase-PKB/Akt signal cascade and the transcription factors PPARγ and C/EBPα in the induction of 3T3-L1 adipocyte differentiation [11,29]. In the present study, DM inhibited DMI treatment-induced phosphorylation of Akt in 3T3-L1 adipocytes. Moreover, treatment of 3T3-L1 cells with a combination of DM and LY294002 showed significantly stronger inhibitory effects on TG accumulation than treatment with DM alone. Therefore, these results indicated that DM strongly caused the suppression of adipogenesis in 3T3-L1 preadipocytes by the downregulation of adipogenic and lipogenic genes through the attenuation of the Akt signaling pathway induced by insulin treatment.

It is well known that inhibition of adipocyte differentiation is related to obesity prevention. Certain herbal and plant extracts, such as catechin, procyanidin, and epigallocatechin gallate (EGCG), have been reported to inhibit adipogenesis [33,34,35]. Rutin and o-coumaric acid also inhibit the expression of PPARγ and C/EBPα, and efficiently suppress adipogenesis in 3T3-L1 adipocytes [36]. Oligonol blocks 3T3-L1 adipocyte differentiation by reducing adipogenic gene expression and inhibiting the phosphorylation of Akt and mammalian target of rapamycin (mTOR) signaling pathway during the early stages of adipogenesis [37]. Alchemilla monticola (ALM) inhibits the early stage of adipogenesis in human adipocytes through the suppression of PI3K/Akt signaling activity [38]. Indeed, several phenolic compounds block the early stages of adipogenesis by regulating the expression of adipogenic genes [33]. Polyphenols such as EGCG, oligonol, and ALM inhibited adipogenesis via down-regulation of Akt signaling in adipocytes.

## 4. Materials and Methods

### 4.1. Preparation of Derhamnosylmaysin (DM)

The dried, centipedegrass plants were ground and extracted three times with using 80% MeOH at room temperature for 3 days, and then they were filtered; the solvent was evaporated under reduced pressure. This concentrated extract was suspended in 10% MeOH and then partitioned in turn with n-hexane EtOAc, and n-BuOH to yield dried n-hexane, EtOAc, n-BuOH, and H_2_O-soluble residues. A portion of the EtOAc-layer was chromatographed on a Toyopearl HW-40 column (4 cm i.d. × 43 cm, coarse grade) with H_2_O containing increasing amounts of MeOH in a stepwise gradient mode and fractioned into seven subfractions CG1-CG7, respectively. The subfraction CG4 was subjected to column chromatography over a YMC GEL ODS AQ 120-50S column (2.5 cm i.d. × 50 cm, particle size 50 μm) with aqueous MeOH to yield pure derhamnosylmaysin (DM) [7,39].

### 4.2. Cell Culture

Cell culture media and supplements were obtained from Sigma-Aldrich (St. Louis, MO, USA). Mouse 3T3-L1 preadipocytes were grown in high-glucose Dulbecco’s modified Eagle medium (DMEM) with 10% calf serum at 37 °C in a humidified atmosphere of 5% CO_2_. The cells were subcultured after reaching a confluence of 80%. The 3T3-L1 fibroblasts were able to differentiate into fat-laden adipocytes in a span of approximately one week upon induction using fetal bovine serum (FBS), dexamethasone (DEX), 3-isobutyl-1-methylxanthine (IBMX), and insulin [11,12]. A mixture (DMI) of insulin, IBMX, and DEX was used to chemically induce the differentiation of 3T3-L1 cells into adipocytes.

To induce adipogenesis, 3T3-L1 cells were cultured until confluent, and 1 day after reaching confluence, called day 0, the culture medium was changed to differentiation/induction medium (DMI) containing 100 mM insulin, 0.5 mM 3-isobutyl-1-methylxanthine, and 0.25 μM dexamethasone in DMEM containing 10% fetal bovine serum. The differentiation/induction medium (DMI) was changed every 2 days. The 3-Isobutyl-1-methylxanthine, dexamethasone, and Oil Red O were obtained from Sigma-Aldrich (St. Louis, MO, USA). DM [7,39] was added to the culture medium containing adipocytes on day 0. Cells were treated with 0, 1.1, or 2.2 µM DM compound. After treatment with DM for 4 or 8 days, the 3T3-L1 adipocytes were lysed for experimental analysis. To analyze cell viability, the cytotoxicity of DM was evaluated by a 3-(4, 5-demethylthiazol-2-yl)-2, 5-diphenyltetrazolium bromide (MTT) assay.

### 4.3. Oil Red O Staining

To investigate the inhibitory effect of DM on lipid accumulation in 3T3-L1 cells, we used Oil Red O staining after adipocyte differentiation. The 3T3-L1 preadipocytes were treated with varying concentrations of DM (0, 1.1, and 2.2 µM) at the beginning of the differentiation process to adipocytes (day 0) to assess the impact of treatment on lipid accumulation. Cells were treated either with DM (1.1 µM or 2.2 µM) or vehicle in the differentiation medium for days 0-8 of adipogenesis. After washing, the cells were stained for 1 hr at room temperature in freshly diluted Oil Red O containing 0.5% Oil Red O in isopropanol. After staining the lipid droplets, the Oil Red O staining solution was removed, and the culture plates were rinsed with water and dried. The lipid droplets in Oil Red O-stained cells were directly visualized and imaged under an Olympus microscope (Olympus, Tokyo, Japan).

### 4.4. Measurement of Triglyceride (TG) Content

Cellular TG contents were measured using a commercial TG assay kit (Sigma-Aldrich, St Louis, MO, USA) according to the manufacturer’s instructions. The cells were incubated with 0, 1.1, and 2.2 μM DM during adipogenesis. During differentiation, the 3T3-L1 cells were incubated with or without the selective PI3K inhibitor LY294002, which was purchased from Sigma-Aldrich (St. Louis, MO, USA) at a concentration of 10 μM in the presence or absence of DM for 8 days. To analyze the content of cellular triglycerides, cells were washed with PBS and then scraped into 200 μL PBS and homogenized by sonication for 1 min. The lysates were assayed for total triglycerides by using the assay kit. The results are expressed as percentage changes. All the experiments were carried out in triplicate.

### 4.5. RT-PCR

Total RNA was isolated from 3T3-L1 adipocytes using TRIzol reagent (Invitrogen, Carlsbad, CA, USA). First-strand cDNA synthesis was performed using oligo (deoxythymidine) primers and SuperScriptII reverse-transcriptase (Invitrogen, CA, USA). The target cDNA was amplified using the following sense and antisense primers: sense 5′-GACTACGCAACACACGTGTAACT-3′ and antisense 5′-CAAAACCAAAAACATCAACAACCC-3′ for C/EBPβ; sense 5′-TTTTCAAGGGTGCCAGTTTC -3′ and antisense 5′-AATCCTTGGCCCTCTGAGAT-3′ for PPARγ; sense 5′-TTACAACAGGCCAGGTTTCC-3′ and antisense 5′-GGCTGGCGACATACAGATCA-3′ for C/EBPα; control detection of β-actin was performed with sense (5′-GACAACGGCTCCGGCATGTGCAAAG-3′) and antisense (5′-TTCACGGTTGGCCTTAGGGTTCAG-3′) primers under the same conditions.

### 4.6. Western Blot Analysis

Western blotting was performed according to standard procedures [15]. Proteins were separated by 10% SDS-polyacrylamide gel electrophoresis and transferred onto a polyvinylidene fluoride membrane (Amersham Pharmacia, UK). Blots were incubated with the primary antibody at 4 °C overnight. PPARγ (1:1000 dilutions), C/EBPβ (1:2000 dilutions), C/EBPα (1:1000 dilutions), SREBP-1c (1:1000 dilutions), ACC (1:1000 dilutions), SCD-1 (1:1000 dilutions), aP2 (1:2000 dilutions), Akt (1:2000 dilutions), and FAS (1:1000 dilutions) antibodies were purchased from Cell Signaling, and a monoclonal β-actin antibody was purchased from Chemicon. HRP-labeled mouse anti-rabbit IgG was purchased from Jackson ImmunoResearch. The chemiluminescence kit was from Pierce (Rockford, IL). Then, blots were washed three times for 10 min with 1X TBS and 0.1% Tween^®^ 20 and incubated for 1 hr with secondary antibody (peroxidase-coupled anti-rabbit in 1X TBS and 0.1% Tween^®^20). After washing three times for 10 min, immunoreactive proteins were detected using a chemiluminescent ECL assay kit (Amersham Pharmacia, UK) according to the manufacturer’s instructions.

### 4.7. Statistical Analysis

Statistical methods Statistical analyses were performed using SPSS (SPSS Inc., Chicago, IL, USA). Data are expressed as means ± SD. Differences among multiple groups were evaluated with a one-way analysis of variance (one-way ANOVA) with Tukey’s multiple comparison test. The differences were considered significant at *p* < 0.05.

## 5. Conclusions

In conclusion, our results suggested that exposure of preadipocytes to DM inhibited adipocyte differentiation and lipid accumulation in 3T3-L1 adipocytes through downregulation of adipocyte transcription factors and lipid accumulation-related genes. These findings strongly indicated that DM significantly suppressed adipogenesis through inhibition of the Akt signaling pathway in 3T3-L1 cells. As far as we know, it is the first report that to characterize the functional aspect of DM as an anti-adipogenic compound in the cellular model, which features inducible differentiation of adipocytes. The key features of molecular signature (i.e., DM-driven anti-adipogenicity molecular network including mRNA and protein expression levels) through these studies could be beneficial to contribute to future design and animal studies exploring the natural compounds taking advantage of structure-dependent activity (SAR) using DM as a hit compound isolated from natural herb and plants. In further studies such as functional metabolomics studies that use the animal experiment, such as diet-induced obesity models and/or transgenic mice models equipped with gene-editing technology, it might be necessary to validate and evaluate the structure-specific lead compound and its targeting molecule, which would be associated with DM as a hit compound, a natural seed, and a potential therapeutic alternative. For future food science revenue, nutraceutical supplements are worth enhancing toward metabolic modulation related to preventing and intervening obesity, in particular, adolescence obesity.

## Figures and Tables

**Figure 1 molecules-27-04232-f001:**
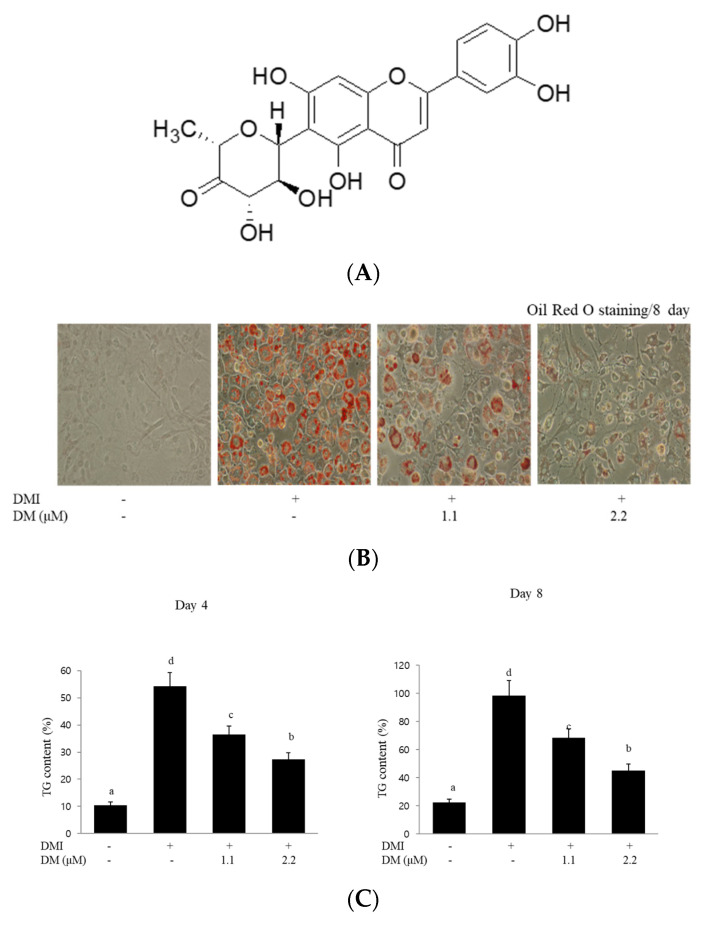
Effect of DM on lipid accumulation, TG content, and cell viability in 3T3-L1 cells. (**A**) Chemical structure of DM. (**B**) Accumulation of intracellular lipid drops in Oil Red O staining of the controls (DMI+, DM−) or DM-treated cells on day 8 following the induction of differentiation. (**C**) Measurement of intracellular TG content in control and DM-treated 3T3-L1 cells. Data are the mean ± SD of three independent experiments, each performed in triplicate. Values labeled with different letters (a–d) are significantly different (*p* < 0.05). Differences among the multiple groups were determined based on a one-way analysis of variants, followed by Tukey’s post hoc test. (**D**) Measurement of cell viability in control or DM-treated 3T3-L1 cells using the MTT assay. Data are the mean ± SD of three independent experiments, each performed in triplicate.

**Figure 2 molecules-27-04232-f002:**
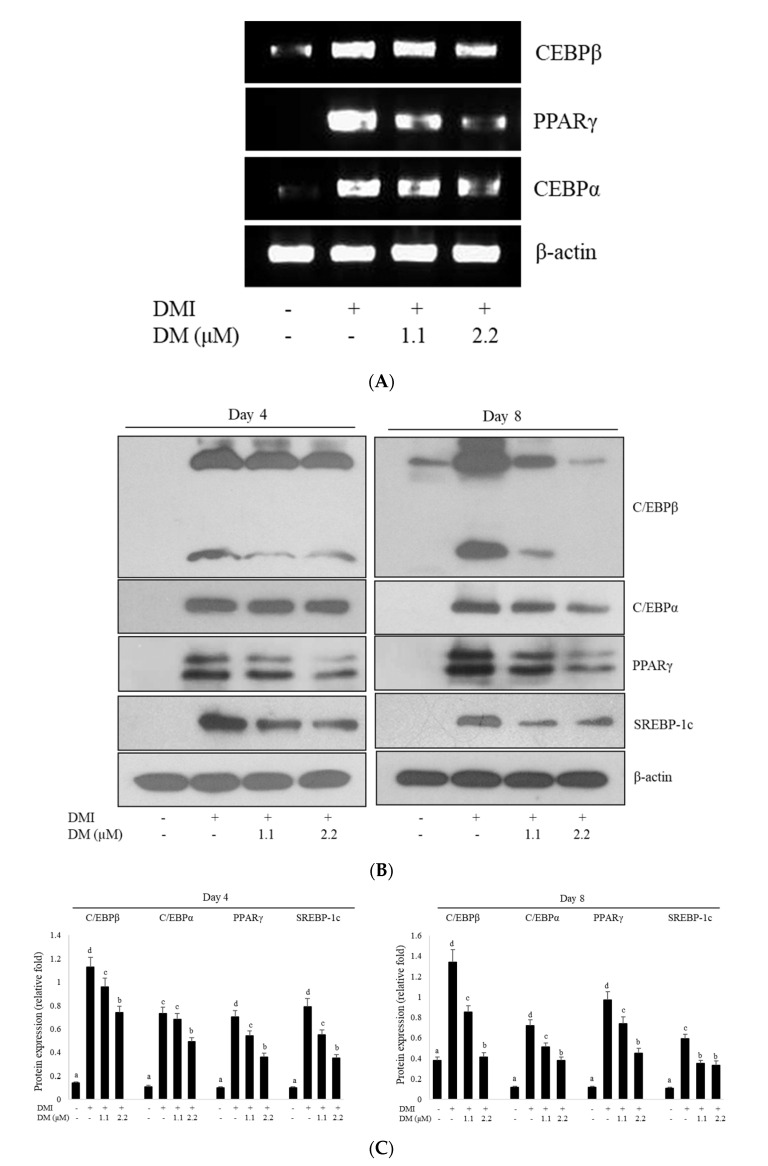
Effect of DM on the mRNA and protein expression levels of adipogenic transcription factors during adipocyte differentiation. (**A**) The expression of C/EBPβ, C/EBPα, and PPARγ was examined by RT-PCR using specific primers on day 4 after induction of adipocyte differentiation. (**B**) DM inhibited the protein expression of transcription factors regulating adipogenesis in 3T3-L1 adipocytes. Western blotting was performed using cell lysates from 3T3-L1 cells prepared on day 4 or day 8 after the induction of differentiation. (**C**) Densitometry analyses of C/EBPβ, C/EBPα, PPARγ, and SREBP-1c were shown. Expression levels of target proteins were assessed by Western blotting with β-actin as an internal control. All values given were mean ± SD of three independent experiments. Values labeled with different letters (a–d) are significantly different (*p* < 0.05).

**Figure 3 molecules-27-04232-f003:**
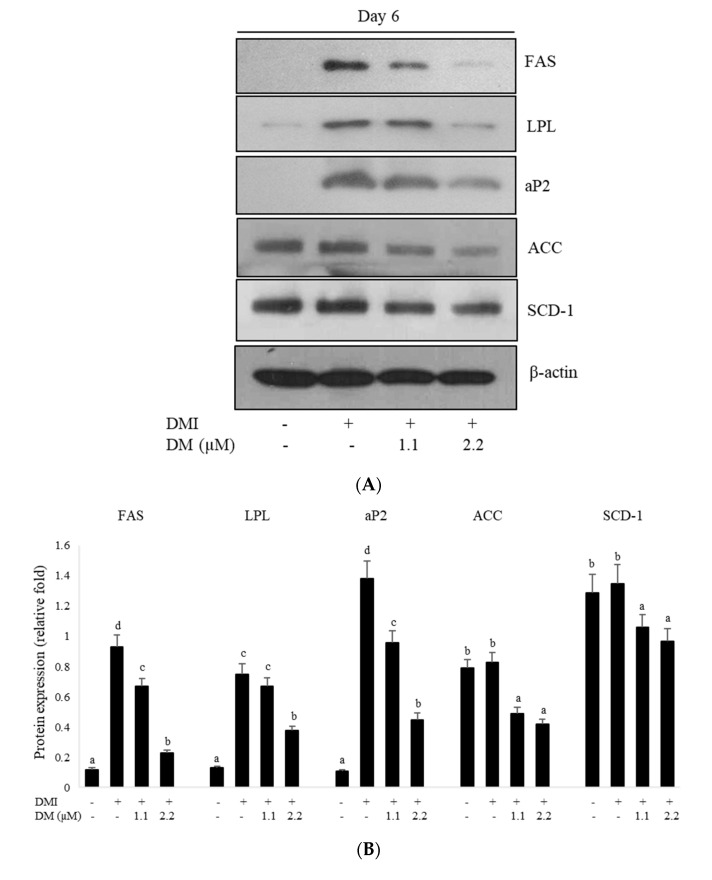
Effect of DM on the expression of adipogenesis-related genes and adipogenesis signaling-related genes. (**A**) The expression of aP2, LPL, FAS, SCD-1, and ACC was assessed during the differentiation of 3T3-L1 cells. (**B**) Relative protein expression level of adipogenesis-related genes. β-actin expression in each sample was used as an internal control to normalize expression. The data were expressed as the mean ± SD of three independent experiments. Values labeled with different letters (a–d) are significantly different (*p* < 0.05). (**C**) Effect of DM on Akt phosphorylation during 3T3-L1 differentiation. DM treatment attenuated the phosphorylation level of Akt at days 4 and 8 after induction of adipocyte differentiation. (**D**) The phosphorylation of Akt was normalized to the total Akt expression level. Values were presented as the mean ± SD of three independent experiments. Different letters within a variable are significantly different (*p* < 0.05). (**E**) Effects of the Akt inhibitor LY294002 on the DM-induced inhibition of adipogenesis in 3T3-L1 cells. To determine whether signaling of lipid accumulation is associated with Akt/PI3K signaling in the 3T3-L1 cells, cells were treated with DM in the presence or absence of LY294002 (10 µM) under the condition of differentiation. The intracellular lipid accumulation was measured using a TG assay. The data were represented as the mean ± SD of three independent experiments. Alphabetical labels with different letters (a–d) are significantly different compared to the control (*p* < 0.05).

## Data Availability

Not applicable.

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
