# Peer review of "Derhamnosylmaysin Inhibits Adipogenesis via Inhibiting Expression of PPARγ and C/EBPα in 3T3-L1 Cells"

_molecules, 2022, doi:10.3390/molecules27134232_

Round 1

Reviewer 1 Report

The manuscript entitled “Derhamnosylmaysin Inhibits Adipogenesis Via Suppressing Expression of PPARγ” and C/EBPα in 3T3-L1 Cells” by Cho et al., describes the effects of DM, a flavonoid contained in centipede grass extracts, on 3T3-L1 preadipocyte differentiation and lipogenesis. The paper adds important knowledge regarding the anti-obesogenic effects of natural compounds. I only have two major points before considering it suitable for publication in Molecules:

-           The title word “suppressing” suggests total elimination of expression of PPARγ and C/EBPα, instead the present results indicate a reduction. I would substitute the word “suppressing” with “inhibiting”

-           The RT-PCR experiments showing a decrease in C/EBPβ, PPARγ and C/EBPα mRNAs were performed with non-quantitative methods.  Therefore, the sentence “could be responsible for almost 55% at the transcriptional level” (line 149) should be removed because not reliable at all.   In order to make a specific quantitative statement you should perform Real-Time RT-PCR. In substitution, you can say that non-quantitative RT-PCR experiments suggest a decrease in C/EBPβ, PPARγ and C/EBPα mRNAs (these experiments cannot define the amount of this decrease).

-           As a consequence, I would leave only the agarose gel picture of the RT-PCR experiment (Fig. 2A) and eliminate the relative bar graph (Fig. 2B).  The subsequent western blotting experiments and densitometric analysis are enough to show a decrease in the expression of C/EBPβ, PPARγ and C/EBPα proteins (Fig. 2C-D, this set of experiments is more reliable) and confirm the previous suggestion made by RT-PCR.

Author Response

Dr. Farid Chemat

Editor-in-Chief

Université d´Avignon et des Pays du Vaucluse, 84029 Avignon, France

Manuscript ID: molecules-1648681

Title: Derhamnosylmaysin Inhibits Adipogenesis Via Suppressing Expression of PPARγ and C/EBPα in 3T3-L1 Cells.

Dear Dr. Farid Chemat

Overall, we appreciate the reviewer’s time and effort to advise their comments to point us to improve the quality of this scientific report following the Molecules. In our humble opinions, this revision satisfies the reviewer’s request following each line of suggestion. Again, thank you for sharing insightful comments encouraging us with future directions as well to improve the quality of our study and further extension utilizing the animal level. We totally agree with that.

Thank you very much for your time and consideration.

Your Sincerely,

Jae-Hyeon Cho, D.V.M. Ph.D

Reviewers' comments:

Reviewer 1

The manuscript entitled “Derhamnosylmaysin Inhibits Adipogenesis Via Suppressing Expression of PPARγ and C/EBPα in 3T3-L1 Cells” by Cho et al., describes the effects of DM, a flavonoid contained in centipede grass extracts, on 3T3-L1 preadipocyte differentiation and lipogenesis. The paper adds important knowledge regarding the anti-obesogenic effects of natural compounds. I only have two major points before considering it suitable for publication in Molecules:

(Q) The title word “suppressing” suggests total elimination of expression of PPARγ and C/EBPα, instead the present results indicate a reduction. I would substitute the word “suppressing” with “inhibiting”.

RESPONSE : We would like to thank the reviewer's comments. The title word in the manuscript is revised.

(Q) The RT-PCR experiments showing a decrease in C/EBPβ, PPARγ and C/EBPα mRNAs were performed with non-quantitative methods. Therefore, the sentence “could be responsible for almost 55% at the transcriptional level” (line 149) should be removed because not reliable at all. In order to make a specific quantitative statement you should perform Real-Time RT-PCR. In substitution, you can say that non-quantitative RT-PCR experiments suggest a decrease in C/EBPβ, PPARγ and C/EBPα mRNAs (these experiments cannot define the amount of this decrease).

RESPONSE : We agree with the reviewer's assessment. The text is revised.

(Q) As a consequence, I would leave only the agarose gel picture of the RT-PCR experiment (Fig. 2A) and eliminate the relative bar graph (Fig. 2B). The subsequent western blotting experiments and densitometric analysis are enough to show a decrease in the expression of C/EBPβ, PPARγ and C/EBPα proteins (Fig. 2C-D, this set of experiments is more reliable) and confirm the previous suggestion made by RT-PCR.

RESPONSE : We appreciate the reviewer's comments. The text is revised according to the comment of the reviewer.

Reviewer 2 Report

In this study, Cho et al present a set of interesting results demonstrating the anti-lipogenic properties of the polyphenol derhamnosylmaysin (DM). The manuscript meets the quality required to be published in Molecules, however, it is necessary to address certain aspects before being accepted. The issues to be addressed are listed below.

1. The authors interpret the effect of the AKT inhibitor as confirmation of the role of this protein in the effects of DM, in my opinion there is a misinterpretation of it. Both DM and LY294002 inhibit MDI-induced TG accumulation, and a cumulative effect (greater reduction) is observed when DM and LY294002 are co-administered. If the effects of AKT and DM overlapped, the cumulative effect would not be observed. Thus, this result should be better interpreted and discussed.

2. The authors should justify the doses of DM used in the study.

3. It would be important to use the term "Protein expression" instead of "Gene expression" when protein analysis was done by western blot (eg Y-axis Fig 2D).

4. The first 3 paragraphs of the discussion are unnecessary, since that information was already given in the introduction. It is better to start with a summary of the results obtained, as the authors did right after.

5. The dilutions of the antibodies must be referred.

6. Line 17, replace "suppressed" with "decreased".

7. Line 169 correct "Densitomery" (Densitometry").

Author Response

Dr. Farid Chemat

Editor-in-Chief

Université d´Avignon et des Pays du Vaucluse, 84029 Avignon, France

Manuscript ID: molecules-1648681

Title: Derhamnosylmaysin Inhibits Adipogenesis Via Suppressing Expression of PPARγ and C/EBPα in 3T3-L1 Cells.

Dear Dr. Farid Chemat

Overall, we appreciate the reviewer’s time and effort to advise their comments to point us to improve the quality of this scientific report following the Molecules. In our humble opinions, this revision satisfies the reviewer’s request following each line of suggestion. Again, thank you for sharing insightful comments encouraging us with future directions as well to improve the quality of our study and further extension utilizing the animal level. We totally agree with that.

Thank you very much for your time and consideration.

Your Sincerely,

Jae-Hyeon Cho, D.V.M. Ph.D

Reviewer 2 Comments,

(Q1). The authors interpret the effect of the AKT inhibitor as confirmation of the role of this protein in the effects of DM, in my opinion, there is a misinterpretation of it. Both DM and LY294002 inhibits MDI-induced TG accumulation, and a cumulative effect (greater reduction) is observed when DM and LY294002 are co-administered. If the effects of AKT and DM overlapped, the cumulative effect would not be observed. Thus, this result should be better interpreted and discussed.

RESPONSE : We appreciate reviewer’s comments. In the present study, we found that DM attenuated the Insulin (DMI)-induced increase in C/EBPs and PPARγ expression and induced a significant decrease in phospho-Akt expression in the 3T3-L1 adipocytes. Many studies have shown that the serine/threonine kinase Akt is particularly important in mediating adipocyte differentiation and the metabolic actions of insulin. Insulin transmits its signal intracellularly by binding to its specific receptor at the cell surface and activating the receptor’s intrinsic tyrosine kinase activity. Insulin signaling activates Akt through PI3K and induces serine/threonine phosphorylation of downstream targeting factors.

In the present study, to explore whether PI3K/Akt signalling pathway is involved in the inhibition of adipocyte differentiation by DM, 3T3-L1 cells were differentiated with or without LY294002 (10 µM) in the presence or absence of the DM for 8 days. LY294002 is chemical inhibitor of PI3K/Akt, which is used extensively to study the role of PI3K/Akt pathway. Incubation of 3T3-L1 adipocytes with LY294002 markedly inhibited the DMI-induced adipocyte differentiation of 3T3-L1 cells. Moreover, combination-treatment of PI3K/Akt inhibitor, LY294002 and DM exhibited more significant inhibitory effect on triglyceride accumulation in 3T3-L1 cells when compare with the LY2904002 alone treatment cells. Thus, triglyceride accumulation was strongly inhibited in the presence of DM, suggesting that DM prevent adipocyte differentiation through an inhibition effect of PI3K/Akt signalling pathway in 3T3-L1 cells.

We partially agreed with our opinion about what the reviewer was concerned about. In the experiment to define the potential signaling pathway, we examined the phenotypically relevant signaling network such as Akt as a pivotal point intracellular level. The limitation of this study, we did not complete functional analysis using siRNA or gene overexpression tools to confirm this verification.

(Q2). The authors should justify the doses of DM used in the study.

RESPONSE : In our study, we try to determine the doses of DM of an effective dose of cellular and molecular dynamics focused clearance dosage on adipogenesis. We examined the effects of drehamnosylmaysin (DM), which is pure and single chemistry, on the differentiation of preadipocytes to adipocytes, 3T3-L1 preadipocytes were treated with various concentrations (0, 1.1 uM, and 2.2 uM) of DM with a DMI mixture (insulin, 3-isobutyl-1-methylxanthine, dexamethasone) for 8 days. Subsequently, the cellular cytotoxicity (viability) induced by different doses of DM was analyzed by MTT assay.

In this study, post-confluent 3T3-L1 preadipocytes were maintained in adipocyte-induction media (DMI) and treated with various doses of DM (0, 1.1 uM, and 2.2 uM). Insulin, IBMX and dexamethasone to chemically induce the differentiation of 3T3-L1 cells into adipocytes, and during the differentiation process (days 0 to 8), cells were treated with DM every day. The treatment of DM concentration 2.2 uM is sufficiently stable during the entire differentiation process of 3T3-L1 cells into adipocytes. Not surprisingly, however, the concentration of above 3.3 uM of DM is toxic to differentiated 3T3- L1 adipocyte, and induce the cell death of 3T3-L1 adipocytes. Thus, concentration range of 1.1 and 2.2 uM DM was appropriate for treatment of cells in the subsequent experiments.

(Q3). It would be important to use the term "Protein expression" instead of "Gene expression" when protein analysis was done by western blot (eg Y-axis Fig 2D).

RESPONSE : We appreciate reviewer’s comments. Fig. 2 D visualized Quantitative data following the explicit effect of DM on the protein expression levels of adipogenic-specific proteins during adipocyte differentiation. Text is revised.

(Q4). The first 3 paragraphs of the discussion are unnecessary since that information was already given in the introduction. It is better to start with a summary of the results obtained, as the authors did right after.

RESPONSE : We appreciate reviewer’s comments. We agreed your comments with direction are very helpful and they need to be revised and rephrased by highlighted color.

(Q5). The dilutions of the antibodies must be referred.

RESPONSE : We agreed that and corrected the description and replace it by adding dilution information (e.g., 1;1,000 dilutions or 1;1,500 dilutions)

(Q6). Line 17, replace "suppressed" with "decreased".

RESPONSE : We appreciate reviewer’s comments and corrected by replacing and rewording that.

(Q7). Line 169 correct "Densitomery" (Densitometry").

RESPONSE : We appreciate reviewer’s comments and corrected it by replacing the typo.

Reviewer 3 Report

This paper has interesting and important information about the influence of derhamnosylmaysin on adipogenesis and lipid accumulation in 3T3-L1 adipocytes. The results show that derhamnosylmaysin may become a potentially beneficial compound for controlling overweight and obesity. However, it should be remembered that these are only tests on cells. It is important to remember that the main cause of obesity is excessive caloric intake. One of the factors preventing obesity is proper supply of calories. It is not known how strong the effect of DM will be when excess calories are supplied to the body. So further research, including research on animals, is very needed.

 However, there are issues that need to be improved.

1. line 45 - Provide examples of other anti-adipogenic compounds (other than polyphenols).

2. Latin names should be written in italics in all of paper, including in vitro or in vivo (line 60, 63…..).

3. Line 120 - Information on dosage and unit of DM is missing.

4. Section 4.1. No information on the origin of DM (who did the extract, who isolated the compound, what method, what control…). This information is important and should be described.

5. Incorrectly formatted references. Make certain that all of the references are formatted properly for Molecules.

Author Response

Dr. Farid Chemat

Editor-in-Chief

Université d´Avignon et des Pays du Vaucluse, 84029 Avignon, France

Manuscript ID: molecules-1648681

Title: Derhamnosylmaysin Inhibits Adipogenesis Via Suppressing Expression of PPARγ and C/EBPα in 3T3-L1 Cells.

Dear Dr. Farid Chemat

Overall, we appreciate the reviewer’s time and effort to advise their comments to point us to improve the quality of this scientific report following the Molecules. In our humble opinions, this revision satisfies the reviewer’s request following each line of suggestion. Again, thank you for sharing insightful comments encouraging us with future directions as well to improve the quality of our study and further extension utilizing the animal level. We totally agree with that.

Thank you very much for your time and consideration.

Your Sincerely,

Jae-Hyeon Cho, D.V.M. Ph.D

Reviewer 3 Comments,

(Q1). line 45 - Provide examples of other anti-adipogenic compounds (other than polyphenols).

RESPONSE : We appreciate the reviewer's comments. Based on the report led by Jakab J et al. (2021), there are listed several potential Phytochemicals from various herbs and plants, and their biological and physiological effect on adipocyte differentiation process by proposing its molecular (or Pharmacokinetic dynamic related to each target (Reference 3. Diabetes Metab Syndr Obes. 2021). Similar to the metabolic disorder including Type 2 diabetes mellitus, and dyslipidemia, we believe the causality of obesity are associated with genetic (e.g., islet defect, abdominal fat, appetite, energy expenditure) and environmental factors (e.g., diet, toxin, physical inactivity, glucose toxicity, more) result from the defect of molecular crosstalk between beta and alpha cells with molecular defect, or cell dysfunction of beta-cell by altering network under the oxidative stress. For example, many studies indicated phytochemicals such as glucoside (c-glycosidic flavone has shown anti-adipogenesis and antioxidant effects (ref. 7) and polyamine (ref. 8). In previous, we tested anti-obesity and adipogenesis using flavonoid extracted from fresh fruits of Rubus crataegifolius Bunge (RCB) in an HFD animal Sprague-Dawley rats model (ref., 15)

(Q2). Latin names should be written in italics in all of paper, including in vitro or in vivo (line 60, 63…..).

RESPONSE : We appreciate the reviewer's comments. Line 62, we corrected 2,2,1-diphenyl-1-picrylhydrazyl (DPPH)-radical scavenging activity. Text is revised as directed.

(Q3). Line 120 - Information on dosage and unit of DM is missing.

RESPONSE : We appreciate the reviewer's comments. We checked DM (µM, micromolar)) used accordingly.

(Q4). Section 4.1. No information on the origin of DM (who did the extract, who isolated the compound, what method, what control…). This information is important and should be described.

RESPONSE : We appreciate the reviewer's comments. The manuscript is revised in the Materials and Methods section.

Preparation of derhamnosylmaysin (DM)

The dried, centipedegrass plants were ground and extracted three times with using 80% MeOH at room temperature for 3 days, and filtered, the solvent was evaporated under reduced pressure. This concentrated extract was suspended in 10% MeOH and then parti-tioned in turn with n-hexane EtOAc, and n-BuOH to yield dried n-hexane, EtOAc, n-BuOH, and H2O-soluble residues. A portion of the EtOAc-layer was chromatographed on a Toyopearl HW-40 column (4 cm i.d. x 43 cm, coarse grade) with H2O containing in-creasing amounts of MeOH in a stepwise gradient mode and fractioned into seven sub-fractions CG1-CG7, respectively. The subfraction CG4 was subjected to column chroma-tography over a YMC GEL ODS AQ 120-50S column (2.5 cm i.d. x 50 cm, particle size 50 μm) with aqueous MeOH to yield pure derhamnosylmaysin (DM) [7, 39]. This report indicated 4.1. Cell culture. Regarding, the DM compound we isolated and characterized the structure and physicochemical identification using analytical chemical tools.

(Q5). Incorrectly formatted references. Make certain that all of the references are formatted properly for Molecules.

RESPONSE : We appreciate the reviewer's comments. We checked all references following the Journal format and the reviewer’s suggestion.

Round 2

Reviewer 3 Report

In my opinion, the article can be published in its current form.

This manuscript is a resubmission of an earlier submission. The following is a list of the peer review reports and author responses from that submission.

Round 1

Reviewer 1 Report

The work by Cho et al described anti-adipogenic features of the treatment with derhamnosylmaysin (DM), a naturally occurring polyphenol from a type of grass, in an in vitro model of differentiating fibroblasts. Although the results appear quite clear, the manuscript fails in the focus of the theoretical background and specially in the results discussions. Thus, my major comments are:

  • Obesity is indeed a complex condition. Despite its definition rests in a positive energy imbalance that ends in fat accumulation; adipogenesis, lipogenesis and lipolysis processes are present in every stage. Thus, adipogenesis is a mechanism that contributes to controlling insulin resistance during adipose tissue expansion, since gather new insulin-sensitive adipocytes. Hence, authors must discuss this aspect in the context of using DM as a tool against adipogenesis.
  • The proposed discussion is far from being complete. Mainly is a repetition of the results. The real discussion comprises the last paragraph, where data from other polyphenols treatments were contrasted. Further discussion regarding type of polyphenol, structure, possible cell interaction, dosage, type of model (cell) used, contrasting literature and the present results is mandatory.
  • Moreover, a detailed discussion regarding possible metabolism of this type of polyphenol is also needed. When DM is consumed, it is expected to be present peripherally? What is the rationale of the used doses of DM?
  • Why an additive effect of DM and akt chemical blockage is indicative of a mechanistical relation between DM and this protein? The additive effect could indicate that further mechanisms, in addition of akt-drive, could be modulated by DM.
  • The mechanisms of adipocyte differentiation have been widely studied before, thus there is no need to be in-depth described. It is not the focus of the manuscript.

Minor comments:

  • English needs major improvement. First paragraphs of the discussion need in-depth re-structuration.
  • “Transcriptional factor” it is not such a specific concept to be added in “keywords”.
  • Revise concept “modulate healthy lifestyle”.
  • Some sections of the introduction can be moved to M&M, also from results.
  • Line 87: PPARy and C/EBPa are also transcription factors (regarding “Transcription factors” of line 86).
  • Resolution of the chemical structure of DM in Figure 1 must be improved.
  • Lines 160-162 are confusing.
  • Line 164: why “major”?.
  • Lines 187-189, must be moved to the end of the legend (works for all figures).
  • Indicate which specific akt phosphorylation was studied.
  • 4 days of 3T3-L1 differentiation seems to be a short period. Can differences between mechanistics modulations be ascribed to different adipogenic stages? Specially regarding late expression molecules.
  • More detail regarding PCR must be included.

Author Response

Reviewer1)

The proposed discussion is far from being complete. Mainly is a repetition of the results. The real discussion comprises the last paragraph, where data from other polyphenols treatments were contrasted. Further discussion regarding type of polyphenol, structure, possible cell interaction, dosage, type of model (cell) used, contrasting literature and the present results is mandatory.

Moreover, a detailed discussion regarding possible metabolism of this type of polyphenol is also needed. When DM is consumed, it is expected to be present peripherally? What is the rationale of the used doses of DM? Thank you.

RESPONSE: We would like to thank the reviewer's comments. Both questions are a very valuable point. it is possible though, that we cannot reach this direction due to a lack of animal study with pharmacokinetic and metabolism studies being needed.

(Q) Why an additive effect of DM and akt chemical blockage is indicative of a mechanistical relation between DM and this protein? The additive effect could indicate that further mechanisms, in addition of akt-drive, could be modulated by DM.

RESPONSE: We appreciate reviewer’s comments. Akt is known to play a major role in adipogenesis and lipid metabolism in insulin signaling mediated-3T3-L1 preadipocytes differentiated into mature adipocytes. A downstream component of insulin signaling (by DMI treatment), the serine/threonine kinase Akt plays a central role in the metabolic actions of insulin and is a marker for insulin signaling. Overexpression of constitutively active Akt in 3T3-L1 adipocytes increased glucose uptake and adipocyte differentiation (Xu and Liao, 2004). A study of Akt-knockout mice showed that Akt is essential for adipocyte differentiation and for the induction of PPARγ expression (Peng et al., 2003). Akt phosphorylates and regulates a large number of substrates that are involved in a diverse array of biological processes, many of which could contribute to the role of Akt in driving adipocyte differentiation.

Many studies have shown that the serine/threonine kinase Akt is particularly important in mediating adipocyte differentiation and the metabolic actions of insulin. Insulin transmits its signal intracellularly by binding to its specific receptor at the cell surface and activating the receptor’s intrinsic tyrosine kinase activity. Insulin signaling activates Akt through PI3K and induces serine/threonine phosphorylation of downstream targeting factors. In the present study, we found that DM attenuated the Insulin (DMI)-induced increase in PPARγ expression and induced a significant decrease in phospho-Akt expression in the 3T3-L1 adipocytes. In addition, other studies also showed that there are several findings related to antiangiogenic and lipid accumulation signaling associated with transcription factors and signaling networks such as oxidative stress-driven-ERK1/2 and PI3K/Akt pathway (Mol Med Rep. 2015 Jul;12(1):1314-20.). Our research cell-based assessment could be engaged in anti-adipogenic with Akt and its alteration of phosphorylation.

(Q) The mechanisms of adipocyte differentiation have been widely studied before, thus there is no need to be in-depth described. It is not the focus of the manuscript.

RESPONSE: We agree with the reviewer's assessment and this is good suggestions for improving the manuscript. However, nobody tested yet natural bioactive compounds like Mode of action in the cellular level utilize DM for obesity. The impact of our study after compiling this research allows for evaluating and exploring novel natural compounds which may contribute to preventing or treatment of the metabolic disorder.

Minor comments:

English needs major improvement. First paragraphs of the discussion need in-depth re-structuration.

RESPONSE: First of all, thank you for sharing your insight and directing us improving our manuscript. The first paragraph of the discussion was edited. Obesity-related public health and health cost concerns due to it triggers secondary metabolic disorders and mental disorders covering a wide range of age spectrum depending on the specific type of disease tropism. Underlying obesity-related metabolic pathogenesis includes adipogenesis and differentiation, adipocytes play an important role in the regulation of energy storage, fatty acid metabolism, and energy expenditure. Adipose tissue also has an important function in energy balance by regulating lipid metabolism and glucose homeostasis [23]. However, excessive fat deposition in adipose tissue is a fast-growing risk for numerous chronic diseases, such as heart disease, diabetes, and hypertension. Overweight-related obesity is no longer regarded as only a cosmetic problem, which is the result of the abnormal fat accumulation associated with an increased risk for the development of numerous adverse health conditions [24, 25].

(Q) “Transcriptional factor” it is not such a specific concept to be added in “keywords”.

RESPONSE: Keywords: exchange “transcriptional factor’ to “lipogenesis”

(Q) Revise concept “modulate healthy lifestyle”.

RESPONSE: We would like to thank the reviewer's comments. Line 51, replaced it modulate healthy metabolic status.

(Q) Some sections of the introduction can be moved to M&M, also from results.

RESPONSE: We appreciate the reviewer's comments. In the present study, we examined the effect of DM on adipocyte differentiation in 3T3-L1 cell model, which was investigated by measuring the accumulation of intracellular droplets of triglyceride as well as the expression levels of several adipogenesis or lipogenesis-related genes. The text is revised in methods section.

(Q) Line 87: PPARy and C/EBPa are also transcription factors (regarding “Transcription factors” of line 86).

RESPONSE: Line 86-88 were restated.

(Q) Resolution of the chemical structure of DM in Figure 1 must be improved.

RESPONSE: We would like to thank the reviewer's comments. The image file is revised.

(Q) Lines 160-162 are confusing.

RESPONSE : Lines 160-165 are rephrased and fixed.

(Q) Line 164: why “major”?.

RESPONSE: Fixed the word as followings: the adipogenic regulator

(Q) Lines 187-189, must be moved to the end of the legend (works for all figures).

RESPONSE: We appreciate reviewer comments. Confirmed those note place to figure legend

(Q) Indicate which specific akt phosphorylation was studied.

RESPONSE: Lysates for 3T3-L1 adipocytes were collected and immunoblotted with total Akt and phospho-Akt (Ser473) antibodies. The text is revised in the text and result section.

(Q) 4 days of 3T3-L1 differentiation seems to be a short period. Can differences between mechanistics modulations be ascribed to different adipogenic stages? Specially regarding late expression molecules.

More detail regarding PCR must be included.

RESPONSE: We appreciate reviewer comments. Adipocyte differentiation is a complex and multi-step process involving a cascade of transcription factors for key proteins that induce gene expression and lead to adipocyte development. As you know, the 3T3-L1 cells, during differentiation, the cells undergo growth arrest and initiate differentiation that is manifested by gene expression and the morphological characteristics of mature adipocytes, such as the accumulation of lipid droplets. The 3T3-L1 cells were fully differentiated by at 7-8 days, and the accumulation of lipids was obviously visualized with Oil red O staining, or measured triglyceride assay. In the our experiments, the development of adipocyte cell phenotype in 3T3-L1 preadipocytes requires approximately 3-4 days, and the preadipocyte cells into mature adipocytes are not reached a maximum status on days 3-4. While after 8 days of inducing adipogenic differentiation, the 3T3-L1 cells are fully or finally differentiated to mature adipocytes and reveals full and well-developed lipid droplets in the intracellular regions.

In general, C/EBPβ is known as the early response gene in adipogenesis. Its expression is transiently enhanced within 1 day after the initiation of adipocyte differentiation, and then decreased. Thus we would better to investigate the expression of the C/EBPβ in the early stage of adipogenesis (within 1 day). However, the reasons we investigate the expression at day 4 differentiation for early gene of C/EBPβ are that 3T3-L1 preadipiocyte have the ability to proliferate and differentiation into mature adipocyte. Expression of the adipocyte phenotype is triggered by treating confluent monolayers with combination of DMI, which chemically induce the differentiation of 3T3-L1 cells into adipocytes. C/EBPβ is expressed in the early stage of adipocyte differentiation and activates the transcription of C/EBPα and PPARγ, and its expression is induced in response to DMI within 1 day of 3T3-L1 cell differentiation.

Thus, even 4 days of culture in differentiation medium, the parts of preadipocytes are not going to be fully differentiated but still proliferation. The development of adipocyte cell phenotype requires approximately 3-4 days and fully differentiated at 7-8 days after DMI induction of adipocyte differentiation of 3T3-L1 cells. Even though C/EBPβ is expressed within 24 hours in the early stage of differentiation, in which stage adipocyte conversion is just initiated to adipogenic program, but not efficient adipose conversion. During 3T3-L1 preadipocyte differentiation, the expression of the adipocyte phynotype results in the expression of numerous adipogenic genes and intracellular triglyceride accumulation. Thus, we have focused to perform the analysis for gene expression at the stage (day 4) of conversion of 3T3-L1preadipocyte into adipocyte.

In the present study we performed to determine the inhibitory activity on DM at day 4 after differentiation of 3T3-L1 preadipocytes into adipocytes, most of all, we would examined the effect of DM to inhibit the transcript expression for transcription factors at day 4 (middle stage) before finally differentiate into fully mature adipocytes.

Reviewer 2 Report

In the paper of H.-H. Cho et al entitled “Effect of derhamnosylmaysin on inhibition of adipogenesis in 3T3-L1 cells” the inhibitory action has been demonstrated of plant (centipede grass)-derived substance, derhamnosylmaysin (DM) on the adipogenic maturation of 3T3-L1 preadipocytes induced by mixture of dexamethasone, 3-isobutyl-1-methylxanthine and insulin.

The inhibitory effect of DM was dose-dependent, observed at early (4 day) and late (9 day) steps of cell differentiation. DM affects cell morphology, triglyceride content and adipogenic marker’s expression, having no effect on cell viability. In addition, authors demonstrate that DM inhibits adipogenesis additively to inhibition of Akt signaling.

The work gives the impression of a completed in vitro study which could be published after some corrections.

Reviewer comments.

1)  The title of the manuscript should be chanced. The phrase “…on inhibition of…” is unnecessary in the title of the manuscript, because DM (as authors demonstrate) is inhibitor by itself.

2)  According to results of this paper, DM inhibits Akt signaling (Fig 3B). At the same time Fig. 3C demonstrates that both DM and Akt inhibitor decrease TG content during cell differentiation. Moreover, effects of DM and LY294002 are summarized if these substances are combined together (Fig. 3C). How authors can explain this phenomenon? Does the effect of DM have some Akt-independent component?

3)  Line 144 (Fig1 legend): What do the authors mean by "control"? Is it the cells cultured with or without DMI?!

4)  Fig. 3C, lines 248-249: Which statistical criteria did you used to calculate the differences between d-c, c-b, b-a?

Stylistic inaccuracies

Lines 59-60:

DM containing its maysin derivatives showed high levels of free radical‑scavenging activity in vitro biochemical assays using α,α-diphenyl-β-picrylhydrazyl (DPPH)‑radical scavenging activity

160-162:

…It was also found that both PPARγ and C/EBPα inhibited the adipogenesis-related transcription factors PPARγ and C/EBPs by almost 55% at the transcriptional level (Fig. 2A and B)…

163-165:

…we investigated the protein expression levels of the major adipocyte C/EBPs and PPARγ during adipocyte differentiation on lipid accumulation in 3T3-L1 adipocytes…

Inscription on the figure 3B: phopho-Akt

Additionally, authors are recommended to proofread the text for further stylistic improvement.

Author Response

Reviewer 2 comments.

1)  The title of the manuscript should be chanced. The phrase “…on inhibition of…” is unnecessary in the title of the manuscript, because DM (as authors demonstrate) is inhibitor by itself.

RESPONSE: Replaced the title as follows; Derhamnosylmaysin Inhibits Adipogenesis Via Suppressing Expression of PPARγ and C/EBPα in 3T3-L1 Cells.

2) According to results of this paper, DM inhibits Akt signaling (Fig 3B). At the same time Fig. 3C demonstrates that both DM and Akt inhibitor decrease TG content during cell differentiation. Moreover, effects of DM and LY294002 are summarized if these substances are combined together (Fig. 3C). How authors can explain this phenomenon? Does the effect of DM have some Akt-independent component?

RESPONSE: We appreciate your sharing your insight. Akt is known to play a major role in adipogenesis and lipid metabolism in insulin signaling mediated-3T3-L1 preadipocytes differentiated into mature adipocytes. A downstream component of insulin signaling (by DMI treatment), the serine/threonine kinase Akt plays a central role in the metabolic actions of insulin and is a marker for insulin signaling. Overexpression of constitutively active Akt in 3T3-L1 adipocytes increased glucose uptake and adipocyte differentiation (Xu and Liao, 2004). A study of Akt-knockout mice showed that Akt is essential for adipocyte differentiation and for the induction of PPARγ expression (Peng et al., 2003). Akt phosphorylates and regulates a large number of substrates that are involved in a diverse array of biological processes, many of which could contribute to the role of Akt in driving adipocyte differentiation. Many studies have shown that the serine/threonine kinase Akt is particularly important in mediating adipocyte differentiation and the metabolic actions of insulin. Insulin transmits its signal intracellularly by binding to its specific receptor at the cell surface and activating the receptor’s intrinsic tyrosine kinase activity. Insulin signaling activates Akt through PI3K and induces serine/threonine phosphorylation of downstream targeting factors. In the present study, we found that DM attenuated the Insulin (DMI)-induced increase in PPARγ expression and induced a significant decrease in phospho-Akt expression in the 3T3-L1 adipocytes.

 It is a very important question for us, it looks like crosstalk between Akt signaling by altering phosphorylation though, we need to do furthermore using animal tissue and explore the key regulator such as microRNAs in anti-adipogenesis following DM mediated metabolic intervention belong to the signaling or not.

3) Line 144 (Fig1 legend): What do the authors mean by "control"? Is it the cells cultured with or without DMI?!

RESPONSE: We appreciate it. Sorry for make a confusion, it is clarified controls (DMI+, DM -) on the day 8

4) Fig. 3C, lines 248-249: Which statistical criteria did you used to calculate the differences between d-c, c-b, b-a?

RESPONSE: We conduct statistical power such as a one-way analysis of variance (one-way ANOVA) with Tukey’s multiple comparison test as mentioned M & M section.

Stylistic inaccuracies

(Q) Lines 59-60:

DM containing its maysin derivatives showed high levels of free radical‑scavenging activity in vitro biochemical assays using α,α-diphenyl-β-picrylhydrazyl (DPPH)‑radical scavenging activity

RESPONSE: DM holds biological functionality by which high levels of free radical scavenging activity in vitro biochemical assays using α,α-diphenyl-β-picrylhydrazyl (DPPH) radical scavenging activity which was derived from structure similarity demonstrated at the bioactive compound, maysin derivative.

(Q) 160-162: It was also found that both PPARγ and C/EBPα inhibited the adipogenesis-related transcription factors PPARγ and C/EBPs by almost 55% at the transcriptional level (Fig. 2A and B)…

RESPONSE: Indeed, DM treatment induced a decrease in the expression of C/EBPβ mRNA. It was also found that mRNA expression of transcriptional factors both PPARγ and C/EBPα inhibited the adipogenesis-related cellular events.

(Q) 163-165:

…we investigated the protein expression levels of the major adipocyte C/EBPs and PPARγ during adipocyte differentiation on lipid accumulation in 3T3-L1 adipocytes…

RESPONSE: Taken together, regulation skewed to adipogenic factors, PPARγ and C/EBPs could be responsible for almost 55% at the transcriptional level (Fig. 2A and 2B),

(Q) Inscription on the figure 3B: phopho-Akt

RESPONSE: We appreciate reviewer’s comments. Corrected Phospho-Akt

(Q) Additionally, authors are recommended to proofread the text for further stylistic improvement.

RESPONSE: Revision was partially conducted by adding Proofreading work

Reviewer 3 Report

The manuscript entitled “Effect of derhamnosylmaysin on inhibition of adipogenesis in 3T3-L1 cells” by Cho et al., describes the effects of DM, a flavonoid contained in centipede grass extracts, on 3T3-L1 preadipocite differentiation and lipogenesis. The paper is interesting and adds important knowledge regarding the anti-obesogenic effects of natural compounds. However, several points should be addressed before considering it suitable for publication in Molecules.

Major comments:

  • The source of derhamnosylmaysin is missing. Please provide details in the Methods section or a reference if described in a previous publication.
  • The study of lipid accumulation was performed with only two different DM concentrations. There should be at least one dose-response with at least 4-5 different concentrations in order to show the range of activity of DM.
  • Cell viability: as for lipid accumulation, there should be a dose response with several different DM concentrations (from nM to high uM). This is especially important to define the range of DM safety.
  • mRNA quantification: what technique was used? how was the graph in figure 2B obtained? There are no details on mRNA quantification in the Method section. From Fig 2A, it seems that simple RT-PCR was employed. If this is the case, I would recommend to remove any statement regarding quantitative estimates, since simple RT-PCR is not considered reliable for quantitation. As an alternative, real-time RT-PCR could be performed.
  • The authors discuss the results of Western blotting in quantitative terms. However, densitometry analysis was not performed or, if performed, not shown.   I would suggest to add this analysis to support quantitative statements.

Minor comments:

  • Line 36: the sentence “extends to serious consequences for health” is the same of “have an adverse effect on health”. Please change it.
  • Line 97: the term “include” is not correct, since aP2, LPL, etc. are not transcription factors, they are transcriptional targets.  Please modify the sentence accordingly.
  • Lines 252-254: this sentence is not clear, please try to improve it
  • Line 285: the sentence “decrease in the lipid droplet number and lipid content in Oil Red O staining” is not totally correct, since the lipid content was not quantified by measuring the ORO staining (ORO was used only to visualize lipid droplets); lipid content was quantified by measuring the triglyceride content with a biochemical assay.  Please modify the sentence accordingly.
  • Lines 305-314: these sentences contain information already presented in the introduction, please reduce this section.
  • Line 479: “4.” was typed twice

Author Response

Reviewer3)

Major comments:

(Q) The source of derhamnosylmaysin is missing. Please provide details in the Methods section or a reference if described in a previous publication.

RESPONSE: We added references that we published previously. We also described the source of DM in the methods.

(Q) The study of lipid accumulation was performed with only two different DM concentrations. There should be at least one dose-response with at least 4-5 different concentrations in order to show the range of activity of DM.

RESPONSE: We appreciate reviewer’s comments. We conducted three experiments following three independent experiments, and the result as final presentation data was visualized. To assess whether DM inhibited the cell viability of 3T3-L1 cells, cells were treated with 0-2 ug/ml DM during differentiation and the cell viability was determined by using the MTT assay. Cell viability was decreased by 2 ug/ml (4.4 uM) DM, while not affected by 0.5 (1.1 uM), 1 (2.2 uM), and 1.5 ug/ml (3.3 uM) DM. Therefore, concentration range of 0.5 - 1.5 ug/ml was appropriate for treatment of cells in the subsequent experiments. In this study, 3T3-L1 preadipocytes were also treated at various concentrations (0, 1.1 uM, and 2.2 uM) of DM for 8 days with differentiation induction.

(Q) Cell viability: as for lipid accumulation, there should be a dose response with several different DM concentrations (from nM to high uM). This is especially important to define the range of DM safety.

mRNA quantification: what technique was used? how was the graph in figure 2B obtained? There are no details on mRNA quantification in the Method section. From Fig 2A, it seems that simple RT-PCR was employed. If this is the case, I would recommend to remove any statement regarding quantitative estimates, since simple RT-PCR is not considered reliable for quantitation. As an alternative, real-time RT-PCR could be performed.

RESPONSE: That’s correct. We applied traditional RT-PCR to determine the expression level of mRNAs which may reflect regulation of angiogenesis in 3T3 L1 cells, it is not exactly matched for semiquantitative analysis though, we measured and converted Image J software to represent a quantitative summary after scanning these bands. In the future, we will adopt the RT-qPCR system to evaluate semiquantitative analysis.

(Q) The authors discuss the results of Western blotting in quantitative terms. However, densitometry analysis was not performed or, if performed, not shown. I would suggest to add this analysis to support quantitative statements.

RESPONSE: We appreciate reviewer’s comments. In the case of protein expression, it was shown the difference following treatment of DM instead of without DM. PPARγ and C/EBPα synergistically activate the down-stream promoters of adipocyte-specific genes such as aP2, LPL, and FAS. SREBP-1c also induces the expression of lipogenic genes, ACC and SCD-1. We focused on the effects of DM in the inhibition of adipogenic and lipogenic genes in mid-late stage of 3T3-L1 differentiation. In the present study, we performed to determine the inhibitory activity on aP2, LPL, ACC, and FAS at day 6 after differentiation of 3T3-L1 preadipocytes into adipocytes. Most of all, we would show derhamnosylmaysin (DM) inhibited the expression of the lipogenic and adipogenic genes at mid-late stage before finally differentiate into fully mature adipocytes.

Minor comments:

(Q) Line 36: the sentence “extends to serious consequences for health” is the same of “have an adverse effect on health”. Please change it.

RESPONSE: We appreciate it. Test is revised as Abnormal or excessive accumulation of body fat extends to serious consequences for health, which may cause metabolic malfunction and interventional effects on different organ health.

(Q) Line 97: the term “include” is not correct, since aP2, LPL, etc. are not transcription factors, they are transcriptional targets.  Please modify the sentence accordingly.

RESPONSE: We appreciate it. These transcription factors regulate normal adipocyte differentiation and targeting molecules such as adipocyte fatty acid-binding protein (aP2), lipoprotein lipase (LPL),

(Q) Lines 252-254: this sentence is not clear, please try to improve it

RESPONSE: We appreciate it. To determine whether signaling of lipid accumulation is associated with Akt/PI3K signaling in the 3T3-L1 cells, cells were treated with DM in the presence or absence of LY294002 (10 µM) under the condition of differentiation. The intracellular lipid accumulation was measured using a TG assay. The data were represented as the mean ± SD of three independent experiments. Alphabetical labels with different letters (a-d) are significantly different compared to the control (p < 0.05).

(Q) Line 285: the sentence “decrease in the lipid droplet number and lipid content in Oil Red O staining” is not totally correct, since the lipid content was not quantified by measuring the ORO staining (ORO was used only to visualize lipid droplets); lipid content was quantified by measuring the triglyceride content with a biochemical assay. Please modify the sentence accordingly.

RESPONSE: Microscopic images of 3T3-L1 cells with Oil Red O staining revealed that the accumula-tion of lipid droplets increased with adipogenic differentiation medium (DMI) treatment. However, our results showed that treatment with DM resulted in a decrease in the lipid droplet spot which may reflect lipid content decrease in Oil Red O staining, indicating that DM inhibited the differentiation of 3T3-L1 preadipocytes by decreasing lipid accu-mulation in the intracellular region.

(Q) Lines 305-314: these sentences contain information already presented in the introduction, please reduce this section.

RESPONSE: We appreciate reviewer’s comments. We slightly modified the part in the text because of important issue to understand the mechanism underlying the anti-adipogenic effect of DM in 3T3-L1 cell study.

(Q) Line 479: “4.” was typed twice

RESPONSE: We appreciate it. Text is revised.

Round 2

Reviewer 3 Report

Reviewer3)

Major comments:

(Q) The source of derhamnosylmaysin is missing. Please provide details in the Methods section or a reference if described in a previous publication.

RESPONSE: We added references that we published previously. We also described the source of DM in the methods.

In the revised version of the manuscript, again I cannot find the source of derhamnosylmaysin used in this work or a reference of previously published work.

(Q) The study of lipid accumulation was performed with only two different DM concentrations. There should be at least one dose-response with at least 4-5 different concentrations in order to show the range of activity of DM.

(Q) Cell viability: as for lipid accumulation, there should be a dose response with several different DM concentrations (from nM to high uM). This is especially important to define the range of DM safety.

RESPONSE: We appreciate reviewer’s comments. We conducted three experiments following three independent experiments, and the result as final presentation data was visualized. To assess whether DM inhibited the cell viability of 3T3-L1 cells, cells were treated with 0-2 ug/ml DM during differentiation and the cell viability was determined by using the MTT assay. Cell viability was decreased by 2 ug/ml (4.4 uM) DM, while not affected by 0.5 (1.1 uM), 1 (2.2 uM), and 1.5 ug/ml (3.3 uM) DM. Therefore, concentration range of 0.5 - 1.5 ug/ml was appropriate for treatment of cells in the subsequent experiments. In this study, 3T3-L1 preadipocytes were also treated at various concentrations (0, 1.1 uM, and 2.2 uM) of DM for 8 days with differentiation induction.

If a cell viability assay was performed with 4 different DM concentrations and the highest induced cytotoxicity, then these data should be fully reported in the article, since this is an important piece of information. The full dose-response for cell viability should be reported before the results on lipid accumulation.  Lack of a cell viability assay with a full dose-response is not acceptable.

(Q) mRNA quantification: what technique was used? how was the graph in figure 2B obtained? There are no details on mRNA quantification in the Method section. From Fig 2A, it seems that simple RT-PCR was employed. If this is the case, I would recommend to remove any statement regarding quantitative estimates, since simple RT-PCR is not considered reliable for quantitation. As an alternative, real-time RT-PCR could be performed.

RESPONSE: That’s correct. We applied traditional RT-PCR to determine the expression level of mRNAs which may reflect regulation of angiogenesis in 3T3 L1 cells, it is not exactly matched for semiquantitative analysis though, we measured and converted Image J software to represent a quantitative summary after scanning these bands. In the future, we will adopt the RT-qPCR system to evaluate semiquantitative analysis.

The method used by the authors (non-quantitative RT-PCR) cannot provide reliable results on mRNA quantitation. For this reason, the graphs (Fig. 2B) and the quantitative statements on mRNA levels should be removed. This is a major problem, since reduction in PPARg and C/EBPa is considered the major finding of this work (see title).     

(Q) The authors discuss the results of Western blotting in quantitative terms. However, densitometry analysis was not performed or, if performed, not shown. I would suggest to add this analysis to support quantitative statements.

RESPONSE: We appreciate reviewer’s comments. In the case of protein expression, it was shown the difference following treatment of DM instead of without DM. PPARγ and C/EBPα synergistically activate the down-stream promoters of adipocyte-specific genes such as aP2, LPL, and FAS. SREBP-1c also induces the expression of lipogenic genes, ACC and SCD-1. We focused on the effects of DM in the inhibition of adipogenic and lipogenic genes in mid-late stage of 3T3-L1 differentiation. In the present study, we performed to determine the inhibitory activity on aP2, LPL, ACC, and FAS at day 6 after differentiation of 3T3-L1 preadipocytes into adipocytes. Most of all, we would show derhamnosylmaysin (DM) inhibited the expression of the lipogenic and adipogenic genes at mid-late stage before finally differentiate into fully mature adipocytes.

The authors are not responding to the question.  I am not questioning on the interpretation of the results, I am saying that, in order to provide quantitative statements, densitometric analysis should be performed on the western blots shown in this article. Since the authors did not use quantitative RT-PCR to measure PPARg, C/EPBa and C/EPBb mRNA levels, there should be at least a quantitative analysis of the corresponding protein levels.